# IgG Charge: Practical and Biological Implications

**DOI:** 10.3390/antib8010024

**Published:** 2019-03-14

**Authors:** Danlin Yang, Rachel Kroe-Barrett, Sanjaya Singh, Thomas Laue

**Affiliations:** 1Biotherapeutics Discovery Research, Boehringer Ingelheim Pharmaceuticals, Inc., Ridgefield, CT 06877, USA; dyang55@its.jnj.com (D.Y.); rachel.kroe-barrett@boehringer-ingelheim.com (R.K.-B.); ssing207@ITS.JNJ.com (S.S.); 2Janssen BioTherapeutics, Janssen Research & Development, LLC, Spring House, PA 19477, USA; 3Department of Molecular, Cellular and Biomedical Sciences, University of New Hampshire, Durham, NH 03861, USA

**Keywords:** analytical electrophoresis, IgG subclasses, monoclonal IgG, protein charge, protein–protein interactions

## Abstract

Practically, IgG charge can contribute significantly to thermodynamic nonideality, and hence to solubility and viscosity. Biologically, IgG charge isomers exhibit differences in clearance and potency. It has been known since the 1930s that all immunoglobulins carry a weak negative charge in physiological solvents. However, there has been no systematic exploration of this fundamental property. Accurate charge measurements have been made using membrane confined electrophoresis in two solvents (pH 5.0 and pH 7.4) on a panel of twelve mAb IgGs, as well as their F(ab’)_2_ and Fc fragments. The following observations were made at pH 5.0: (1) the measured charge differs from the calculated charge by ~40 for the intact IgGs, and by ~20 for the Fcs; (2) the intact IgG charge depends on both Fv and Fc sequences, but does not equal the sum of the F(ab)’_2_ and Fc charge; (3) the Fc charge is consistent within a class. In phosphate buffered saline, pH 7.4: (1) the intact IgG charges ranged from 0 to −13; (2) the F(ab’)_2_ fragments are nearly neutral for IgG1s and IgG2s, and about −5 for some of the IgG4s; (3) all Fc fragments are weakly anionic, with IgG1 < IgG2 < IgG4; (4) the charge on the intact IgGs does not equal the sum of the F(ab’)_2_ and Fc charge. In no case is the calculated charge, based solely on H^+^ binding, remotely close to the measured charge. Some mAbs carried a charge in physiological salt that was outside the range observed for serum-purified human poly IgG. To best match physiological properties, a therapeutic mAb should have a measured charge that falls within the range observed for serum-derived human IgGs. A thermodynamically rigorous, concentration-dependent protein–protein interaction parameter is introduced. Based on readily measured properties, interaction curves may be generated to aid in the selection of proteins and solvent conditions. Example curves are provided.

## 1. Introduction

It is known that charge and charge distribution are important contributors to protein solubility and solution viscosity [1,2,3,4,5,6,7,8,9,10,11]. In general, increased charge correlates with higher solubility and lower viscosity because charge–charge repulsion weakens protein–protein interactions [12]. Experimentally, nonideality is quantified by the thermodynamic second virial coefficient (B_22_ or A_2_), with B_22_ > 0 corresponding to net repulsion and B_22_ < 0 corresponding to net attraction between molecules. Molecules possessing the same sign net charge will repel, while those having opposite charge will attract.

However, net charge alone does not fully capture the effects of charge on B_22_. In particular, dipole moments resulting from asymmetric charge distributions can lead to orientation-dependent protein–protein attraction due to charge–dipole and dipole–dipole interactions, which decrease B_22_ [5,9]. If B_22_ is < 0, highly viscous [5,7,8,9] or opalescent [2] solutions may result at high protein concentrations. Recent work suggests that there may be weak, promiscuous attractive interactions between IgGs [13,14]. These attractive interactions may or may not be entirely electrostatic in origin (e.g., weak hydrophobic interactions could contribute), though the salt and temperature dependence suggest electrostatic attractions are involved. Regardless of their origin, it has been suggested that the weak attraction (apparent monomer–dimer Kds of 10^−4^–10^−3^ M [13,14,15]) may reflect the cooperative free energy needed for effector functions [14].

In addition to the importance of charge in the development of high concentration therapeutic formulations, mAb charge may influence in vivo processes. For example, neonatal Fc receptor (FcRn)-independent clearance rates are lower for mAbs with lower pI values than those with higher pI values [16,17,18], presumably due to decreased nonspecific cell surface binding [16,17,19,20]. Furthermore, basic charge variants of mAbs display stronger binding to the FcγRIIIa receptor and increased antibody-dependent cellular cytotoxicity response compared to more acidic charge variants [21,22]. Finally, there is an increasing body of evidence suggesting that IgG sialylation may impact therapeutic efficacy [23] and IgG function [24]. Together these in vivo and in vitro data show that mAb charge correlates with physical and biological consequences and highlight the need to understand what governs IgG charge.

The majority of biotherapeutic mAbs exhibit pIs ≥ 8 [25], and carry a positive charge under formulation conditions (typically pH 5–6) [2,3,4]. However, it has been known for over 80 years that all serum proteins, including the immunoglobulins, carry a net negative charge under physiological conditions [26]. Furthermore, IgGs from several species are anionic in the pH 5–6 range [27,28]. More recently, it was shown that freshly prepared human polyclonal IgGs have a Debye–Hückel–Henry charge, *Z_DHH_* [26], between −3 and −9 [14]. The narrow range of charge is somewhat surprising since isoelectric focusing analysis of the same sample yielded pIs covering the pH range from less than 4 to greater than 10 [14]. There is no published charge data for mAbs in physiological solvents. Consequently, it is not known whether the charge on therapeutic mAbs falls into the rather narrow range observed for normal human IgGs. It is apparent that a systematic analysis of the charge on mAbs would be useful.

Presented here are charge measurements on twelve anti IL-13 IgGs. Using membrane confined electrophoresis, MCE, data have been acquired for three IgGs, mAb 1, mAb 2, and mAb 3, that bind to different IL-13 epitopes [14]. For each mAb, *Z_DHH_* has been measured for four subclasses, IgG1, IgG2, IgG4, and IgG4Pro. Furthermore, the charge on the Fc and F(ab’)_2_ fragments was measured to determine whether the intact IgG charge is the sum of the Fc and F(ab’)_2_ fragment charges, and to assess how the charge is distributed over the IgG structure. Finally, the charge on the IgGs and their fragments were measured at both pH 5.0 and pH 7.4 to determine how the charge varies between formulation and physiological conditions. The results illustrate how little is known about protein charge and demonstrates the power of analytical electrophoresis in assessing this fundamental property.

### Theoretical Basics of Protein Charge

Protein charge contributes significantly to a variety of biochemical, biophysical, and biological phenomena [29]. Thermodynamically, charge is a system property that depends on temperature, pressure, salt concentration, salt type, and pH [12]. At present there is no way to calculate protein charge accurately. However, charge may be measured with both precision and accuracy [26,30,31]. Of the measurement methods, membrane confined electrophoresis [32,33] is the most accurate and flexible [26,34].

There are a variety of charge descriptions (e.g., *ζ* potential, *Z_effective_*, *Z_DHH_*) [26]. While each description is useful, here we will use *Z_DHH_*, which is the unitless valence resulting from the ratio of the protein charge (in coulombs) to the proton unit charge (e.g., Ca^2+^ has a valence of +2, Cl^−^ has a valence of −1). Calculation of *Z_DHH_* from the free-boundary electrophoretic mobility removes the effects of electrophoresis and the Debye-Hückel solvent ion cloud [26,32,35]. Thus, *Z_DHH_* reflects any changes in protein charge that accompany changes in solvent pH, salt type or salt concentration [26].

Even though proton binding to proteins has been studied extensively [36,37], it has been difficult to reconcile values calculated from amino acid side chain pKas with measurements [5,38,39]. Shifts in the pKas due to net protein charge, proximity of charged residues and protein flexibility are known to occur [36,37,40,41,42,43]. Though H^+^ binding contributes to protein charge, *Z_DHH_* reflects binding by all solvent ions (e.g., Na^+^, PO_4_^2^, Cl^−^) and not just H^+^. It has been known for over 60 years that proteins bind monovalent ions, and bind anions to a greater extent than cations [12,44]. Two non-exclusive models have emerged for the mechanism of anion binding. One model focuses on the tendency for anions to accumulate preferentially at hydrophobic surfaces [38,45]. Based on NMR data, the other model suggests that anion binding may involve amide protons [46].

Because ion binding and dissociation occur rapidly, *Z_DHH_* values are time averages. The extent of fluctuation about the mean value is proportional to the change in charge with ion chemical potential (i.e., the slope of the curve of Z versus log[X]) [36]. If the titration curve is flat (i.e., dZ/dlog[X] ~ 0), there will be very little charge variation, and the charge distribution about the average value will be narrow. A steep titration curve, however, indicates large charge variations which, particularly if they swing around neutrality, result in the inter-molecular attractions that reduce solubility and cause higher viscosities. Thus, measurement of *Z_DHH_* as a function of solvent ion concentration (including pH) may be helpful in finding solvent conditions that optimize solubility and viscosity.

## 2. Materials and Methods

### 2.1. Monoclonal and Human Serum IgGs

Twelve anti-IL13 IgGs comprising three unique variable regions, each constructed as four human IgG subclasses, IgG1, IgG2, wild-type IgG4(Ser^222^), and a hinge mutant IgG4(Pro^222^), were made from stable NS0 cell line at Boehringer Ingelheim. Human serum derived from male AB plasma was purchased from Sigma-Aldrich, St. Louis, MO, USA (cat# H4522). The IgGs were purified by ÄKTA affinity chromatography system and MabSelect Sure resin (GE Healthcare, Chicago, IL, USA) following standard methods [47]. The quality of the purified mAb IgGs and their fragments generated by subsequent enzymatic digestion was evaluated by analytical size-exclusion ultra-performance liquid chromatography (SE-UPLC) using a BEH200 column on the Waters Acquity UPLC system (Waters Corporation, Milford, MA, USA). The mobile phase buffer consisted of 50 mM sodium phosphate (pH 6.8), 200 mM arginine, and 0.05% sodium azide. For each sample run, 10 µg of material was injected onto the column with the running flow rate at 0.5 mL/min for 5 min.

### 2.2. IgG Fragmentation

A FragIT kit with individual spin columns containing the active IdeS, a cycstein protease secreted by Streptococcus pyogenes covalently coupled to agarose beads was used (Genovis, cat# A2-FR2-025). After the IgG sample was buffer exchanged into the cleavage buffer (10 mM sodium phosphate, 150 mM NaCl) and the column was equilibrated with the cleavage buffer, the IgG-enzyme mixture was incubated at 37 °C for an hour on an orbital shaker. The digested fragments were separated from the immobilized enzyme, followed by the purification of F(ab’)_2_ using a supplied CaptureSelect column containing Fc affinity matrix (Thermo Fisher Scientific, Waltham, MA, USA). Upon the collection of the F(ab’)_2_ in the flow-through, the Fc was eluted using the 0.1 M glycine (pH 3.0) elution buffer and immediately neutralized by adding 10% *v*/*v* of 1 M Tris (pH 8.0).

### 2.3. Sample Preparation

Each sample was dialyzed into desired buffers at 4–10 °C overnight using Zeba desalting columns (Thermo Fischer), after which the concentration was determined using appropriate extinction coefficients in NanoDrop™ 8000 Spectrophotometer (Thermo Fischer). Two solvents were used: 10 mM sodium acetate, 50 mM NaCl, pH 5.0; and Dulbecco’s PBS (pH 7.4) containing 8 mM sodium phosphate dibasic, 1.5 mM potassium phosphate monobasic, 2.7 mM KCl, and 138 mM NaCl. The acetate buffer was prepared by diluting chemicals purchased from Sigma into distilled deionized water from a Milli-Q Plus filtration system (Millipore, Burlington, MA, USA) and titrating to the desired pH 5.0 with 10 N NaOH solution. For all measurements, the sample solutions were used within a week of preparation and stored at 4 °C between measurements.

### 2.4. Liquid Chromatography Mass Spectrometry (LC-MS)

The sequences of the purified mAbs and respective F(ab’)_2_ and Fc fragments were evaluated by LC-MS using a PoroShell 300SB-C8 column (5 µm, 75 × 1.0 mm) on the Agilent HPLC system followed by analysis in the Agilent 6210 time-of-flight mass spectrometer (Agilent Technologies). The composition of the mobile phase A was 99% water, 1% acetonitrile, and 0.1% formic acid, and that of mobile phase B was 95% acetonitrile, 5% water, and 0.1% formic acid. The gradient started with 20% B at 0 min and increased to 85% B at 10 min with the constant flow rate of 50 µL/min. Each sample was subjected to a native run, a reduced run after incubation with TCEP (Sigma), and a deglycosylated run after incubation with TCEP and PNGase F (New England Biolabs). The MassHunter Qualitative Analysis program (version B.06.00, Agilent, Santa Clara, CA, USA) was used to deconvolute the raw data.

### 2.5. Analytical Ultracentrifugation (AUC)

The solution properties of the purified mAbs and cleaved F(ab’)_2_ and Fc were evaluated by sedimentation velocity experiments in an Optima XL-I AUC equipped with absorbance optics (Beckman Coulter, Brea, CA, USA). Each sample was prepared in three concentrations with 1:3 serial dilutions starting from 0.5 mg/mL in the corresponding buffer, and 400 μL of the prepared solution was loaded into the sample chamber, whereas buffer was loaded into the reference chamber of an AUC cell assembled with standard double-sector centerpieces and quartz windows. The experiments were conducted at 20 °C using an An60Ti 4-hole rotor spinning at 40,000 rpm. The sedimentation process was monitored by collecting absorbance data at 280 nm wavelength and 30-µm radial increments. The collected data was analyzed using the SEDANAL software by which the apparent sedimentation coefficient distribution g(s*) was derived [48]. The resulting analysis was initially plotted as g(s*) vs. s* in which the areas under the peaks provided the concentration for the boundary corresponding to each peak in the distribution. The weight average sedimentation coefficient (*s_w_*) was computed by selecting a range over which to do the average on the plots. The plots were concentration-normalized to enable the inspection for reversible interactions. The Stokes radius, *R_s_*, which is used for *Z_DHH_* calculation is derived from the Svedberg equation:(1)$$R_{s}=\frac{M\left(1-\bar{v}\rho \right)}{sN_{A}6\pi \eta }$$
where *M* is the molar mass, ῡ is the partial specific volume, ρ is the solvent density, *s* is the sedimentation coefficient, *N_A_* is the Avogadro’s number, and *η* is the viscosity of the solvent.

### 2.6. Imaged Capillary Isoelectric Focusing (icIEF)

The pI and charge heterogeneity of the IgG samples were determined on an iCE3 system (Protein Simple) [49,50]. Briefly, the pH gradient was created by an ampholyte mixture consisted of 44% (*v*/*v*) of 1% methylcellulose, 1.25% (*v*/*v*) of pharmalyte 3–10 solution, 3.75% (*v*/*v*) µL of pharmalyte 5–8 solution, 1.25% (*v*/*v*) of servalyte 9–11 solution, 0.63% (*v*/*v*) of pI marker pH 6.14, 0.63% (*v*/*v*) of pI marker pH 8.79, 6.3% (*v*/*v*) of 200 mM iminodiacetic acid, and 43% (*v*/*v*) of water. After sample preparation at 1 mg/mL in DI water, 40 µL of the diluted sample was mixed with 160 µL of ampholyte mixture and centrifuged for 5 min. The operating protocol used an initial potential of 1500 volts for 1 min, followed by a potential of 3000 volts for 20 min. For samples containing highly basic species, pI markers at pH 7.55 and pH 9.77 (0.63% *v*/*v*) and a focus period of 10 min at 3000 volts was used. Separation was monitored at 280 nm, and the data analyzed using the iCE CFR software to calibrate the pI values and to select the markers. Subsequently, the data files were exported to Empower for analysis using the cIEF processing method.

### 2.7. Membrane-Confined Electrophoresis (MCE) and *Z_DHH_* Determinations

Protein valence was measured in the MCE instrument (Spin Analytical, Inc., Berwick, Me, USA), which provides a direct measurement of the electrophoretic mobility (*µ*) to derive the *Z_eff_* and the *Z_DHH_* [32,33]. In each experiment, 20 µL of sample at 1 mg/mL was loaded into a 2 *×* 2 *×* 4 mm quartz cuvette whose ends were sealed with semipermeable membranes (MWCO 3 kDa, Spectra/Por Biotech grade). An electric field was applied (4.3 V/cm for IgG, 8.5 V/cm for F(ab’)_2_ and Fc, and 19.8 V/cm for serum IgGs) longitudinally across the cell. The applied electric field, *E*, is a function of the applied current, *i*, the buffer conductivity (*κ*, 5.8 mS for 10 mM acetate, 50 mM NaCl [pH 5.0] and 16.8 mS for PBS [pH 7.4]), and the cross-sectional area of the cuvette, *A*, as $E=\frac{i}{\kappa A}$. Image scans of the cuvette were acquired with 25 μm resolution at 280 nm every 10–20 s. Time difference analysis provided an apparent electrophoretic mobility distribution, g(μ) versus μ, uncorrected for diffusion. Values of μ were converted to charge using the Spin Analytical software:(2)$$Z_{eff}=\frac{\mu }{fe}$$
(3)$$Z_{DHH}=Z_{eff}\frac{1+\kappa_{D}a}{H\left(\kappa_{D}a\right)}$$
where *µ* is the electrophoretic mobility, *f* is the translational frictional coefficient, *e* is the elementary proton charge, *ĸ_D_* is the inverse Debye length, *a* is the sum of the Stokes radius of the macromolecule and its counterion (0.18 nm for Cl^−^^1^ and 0.122 nm for Na^+^), and *H(κ_D_ a)* is Henry’s function that accounts for electrophoretic effects. For reference, under the experimental conditions used here, *κ_D_a* ~ 2 and **H(*κ_D_a*) ~ 1.1, though exact values are calculated for each experiment.

### 2.8. Calculated Charge, *Z_cal_*, and Calculated Isoelectric pH, pI_Cal_

Sednterp was used to calculate pI values, pI_Cal_, as well as the H^+^ titration curve from which *Z_cal_* was determined [51]. These calculations are based on the amino acid composition and use pKa values from Edsall and Wyman [52]. It was assumed that the N-terminal amino groups were not blocked.

### 2.9. Dynamic Light Scattering (DLS) and k_D_ Determinations

A DynaPro Plate Reader (Wyatt Technology, Santa Barbara, CA, USA) running Dynamics (version 7.4.0.72) was used to determine the diffusion interaction parameter, *k_D_*. Each sample was prepared at 5 concentrations ranging from 10 mg/mL to 0.625 mg/mL in 2-fold serial dilutions. 35 μL of each solution was added to a 384-well UV-Star Clear Microplate (Greiner Bio-One), spun in a centrifuge for 2 min to remove air bubbles and then placed into the plate reader. The experiment was started after the temperature inside the reader reached 20 °C. A total of 10 acquisitions at 20 s per acquisition were obtained for each sample. A well image was acquired after the last acquisition measurement to look for bubbles or deposited aggregates. The mutual diffusion coefficient (*D_m_*) was plotted against the sample concentration *D_m_* = *D*_0_(1 + *k_D_C*), with *D*_0_ and *k_D_* determined by linear regression analysis using GraphPad Prism (version 7.03). The error for *k_D_* was determined by calculating the propagation of the standard error of the coefficients from the linear regression.

### 2.10. Calculation of the Protein–Protein Interaction Curve

Thermodynamic nonideality reflects a balance of repulsive (*B*_22_ > 0) and attractive interactions (*B*_22_ < 0) between molecules. Only two protein characteristics contribute to positive *B*_22_ values, charge–charge repulsion (when the molecules have the same sign charge) and excluded volume (always repulsive). The contribution charge–charge repulsion, including the impact of the Debye-Hückel counterion cloud, may be calculated from $B_{Z}=\frac{1000Z_{2}^{2}{\bar{v}}_{1}}{4m_{3}M_{2}^{2}}\left(\frac{1+2\kappa r_{s}}{{\left(1+\kappa r_{s}\right)}^{2}}\right)$, where $Z_{2}^{2}$ is the square of the protein charge (i.e., *Z_DHH_*), ${\bar{v}}_{1}$ the solvent partial specific volume (mL/g), *M*_2_ the protein molecular weight (g/mol), *m*_3_ the salt molality (mol/kg), *κ* the inverse Debye length (cm^−1^) and *r_s_* the solvated protein radius (cm) [12,44]. The excluded volume includes contributions from the shape of the molecule (in this case, using the axial ratio) and the hydration layer [12,44]. The excluded volume contribution is $B_{Ex}=\frac{8VN_{A}}{2M_{2}{}^{2}}$, where *V* is the solvated protein volume (mL/particle) and *N_A_* is Avogadro’s number [12,44]. The overall repulsive nonideality, B_22_, is the sum of these two contributions. The weak attractive interactions observed for IgGs may be expressed in terms of a dimer dissociation, e.g., $IgG_{2}↔IgG+IgG$, with the strength given by the dissociation constant given by $K_{d}=\frac{{\left[IgG\right]}^{2}}{\left[IgG_{2}\right]}$. At any concentration, the weight-average molecular weight, *M_w_*, of a monomer–dimer mixture may be calculated knowing *K_d_*. For systems exhibiting only repulsive interactions, the slope of a graph of 1/*M_w_* versus concentration, *C* (in g/mL), *B*_22_ (in ml-mole/g^2^), will be positive. Often, a graph of *M*_1_/*M_w_*, where *M*_1_ is the monomer molar mass, is used to ‘normalize’ the data, in which case the slope of the line is *M*_1_∙*B*_22_ and is called *A*_2_ (in mL/g) in the literature. In either case, for purely repulsive systems over a wide concentration range, *B*_22_ or *A*_2_ are positive and constant. For a system that exhibits a mass-action association, *M_w_* increases with concentration (1/*M_w_* or *M*_1_/*M_w_* decrease, producing a negative curve). However, even in the face of self-association, *B*_22_ (or *A*_2_) remain constant and positive, and push the curve in the opposite direction of self-association. Thus, for systems exhibiting both repulsive nonideality and weak self-association, unusual curves may result, starting with a negative slope at low *C* and winding up with a positive slope at higher *C*. The slope of the 1/*M_w_* or *M*_1_/*M_w_* curve at each concentration, then, is an apparent *B*_22_, *B*_22*-app*_, or *A*_2_, *A*_2*-app*_. It is important to note that both *B*_22*-app*_ and *A*_2*-app*_ are thermodynamic parameters and represent useful protein–protein interaction parameters. For this work, data are presented as *A*_2*-app*_. *A*_2*-app*_ is > 0 for net repulsion, <0 for net attraction and =0 for a thermodynamically ideal system.

## 3. Results

### 3.1. Solution Properties of IgGs and Their Fragments

All purified IgGs were subjected to purity characterization by SE-UPLC, sequence identify and glycoform distribution analysis by LC-MS. As summarized in Table 1, all purified materials contain >99% monomer content and were confirmed by sequence to be in the expected IgG subclass with typical distribution of G0F, G1F, and G2F asparagine (N)-linked glycoforms.

The IgGs also displayed homogeneous solution properties within each mAb group in either pH 5.0 acetate and pH 7.4 PBS as illustrated in Figure 1 by the overlapping g(s*) curves. The weight-average sedimentation coefficients (*s_w_*) are mAb1 6.37 ± 0.06, mAb2 6.37 ± 0.05, and mAb3 6.43 ± 0.09 in pH 5.0 acetate, and mAb1 6.28 ± 0.04, mAb2 6.27 ± 0.07, and mAb3 6.31 ± 0.06 in pH 7.4 PBS. These *s_w_* values are consistent with the molecular weight of ~150 kDa IgG antibodies.

IgG cleavage sites and fragment purity are presented in Table 2. The solution homogeneity of each cleaved fragment was assessed by SV-AUC. All IgG fragments showed sedimentation distribution profiles like that in Figure 2 for mAb 1, where the superposition of the three concentrations of F(ab’)_2_ and Fc samples indicate homogeneity and the absence of self-association. The weight-average sedimentation coefficients (*s_w_*) from the Fc evaluations are 3.45 ± 0.02, 3.46 ± 0.02, and 3.38 ± 0.18 for IgG1, IgG2, and IgG4/IgG4Pro, respectively. These values are consistent with the molecular weight of ~50 kDa, which indicates the Fc fragment remains a homodimer in solution despite cleavage below the hinge region. The *s_w_* from the F(ab’)_2_ evaluations are 4.86 ± 0.01, 5.14 ± 0.06, 4.90 ± 0.02, and 4.95 ± 0.01 for IgG1, IgG2, IgG4, and IgG4Pro, respectively. These values are consistent with the molecular weight of ~100 kDa, which is expected for a bivalent Fab.

### 3.2. Isoelectric Point and Correlation to Calculated Values

All IgGs exhibited pI profiles similar to that in Figure 3 for mAb 1 IgG1. Three-peaks are observed, acidic, main and basic. The pI values for each IgG are presented in Table 3, along with the calculated pI. For each mAb, the subclass pIs followed the trend: IgG1 > IgG2 > IgG4, with those of IgG4 and IgG4Pro being identical. The measured main species pI and the calculated pI are correlated (Figure 4), though the intercept (−1) suggests that pI_Cal_ corresponds to the more acidic species.

### 3.3. Net Charge of IgGs and Fragments in Formulation and Physiological Solutions

Using MCE, the electrophoretic mobility was determined for each IgG and its cleaved F(ab’)_2_ and Fc in pH 5.0 acetate and pH 7.4 PBS as illustrated in Figure 5. By applying the Debye–Hückel approximation to correct for the solvent shielding effects, Henry’s function to correct for electrophoretic effects, and using the sum of the measured protein Stokes radius and its counterion, the *Z_DHH_* distribution may be calculated from the electrophoretic mobility (Figure 5, right-hand panels).

Table 4 and Table 5 summarize the *Z_DHH_* measurements, as well as the calculated charge, *Z_cal_*, in pH 5.0 acetate and pH 7.4 PBS, respectively. A 0 charge was assigned if no boundary formed during electrophoresis regardless of the E field direction or magnitude. In acetate pH 5.0 all IgGs and their fragments are cationic (Table 4). However, in all cases the measured *Z_DHH_* is substantially lower than *Z_cal_*. In PBS pH 7.4 (Table 5), all intact IgGs are neutral (mAb 2/IgG1) or anionic, despite the fact the *Z_cal_* is cationic in some cases. For all mAbs, *Z_DHH_* decreases with subclass in the rank order of IgG1 > IgG2 > IgG4.

While *Z_DHH_* and *Z_cal_* are correlated in either solvent (Figure 6), the slope is about 1/2–3/4 of what would be expected if there were a 1:1 correspondence between the expected H^+^ uptake/release and *Z_DHH_*. These data are consistent with a model in which an anion is bound for every 1.3–2 H^+^ bound. Similarly, *Z_DHH_* for the intact IgGs correlates with the sum of *Z_DHH_* from fragments (Figure 7), albeit with a slope that is about ½ of that expected if the charge on the fragments simply summed. We have no mechanism or explanation for the data in Figure 7 and present them here in the hope that they will encourage future work.

## 4. Discussion

Protein charge is a fundamental property that directly influences its structure, stability, solubility, and ability to interact with other macromolecules [53]. Charge–charge repulsion is important for overcoming the attractive forces that lead to high viscosities in high-concentration protein solutions [54]. Because protein charge can vary with solvent conditions, it is a *system* property rather than a property of the protein. The systematic analysis of twelve mAbs and their F(ab’)_2_ and Fc fragments provides several insights into IgG charge and raises several fundamental questions about our understanding of protein charge.

Charge–charge repulsion contributes to thermodynamic nonideality and, consequently, the colloidal stability of protein solutions [12]. It is clear from the data in Table 4 and Table 5 that charge calculations based solely on H^+^ binding lead to highly inaccurate estimates of IgG charge. Thus, even though there is a correlation between the measured and calculated charge (Figure 6), charge calculations should not be considered reliable. Given its potential importance to colloidal stability, it is important to determine the impact of charge on nonideality.

At low to moderate protein concentrations (<~15 mg/mL), the net sum of all repulsive and attractive interactions is described by the second virial coefficient, *B*_22_ or *A*_2_. The diffusion interaction parameter, *k_D_*, is related to and often used as a stand-in for these quantities [55], with more positive values of *k_D_* correlating with more positive values of *B*_22_, i.e., greater repulsive interactions. If charge–charge repulsion contributes significantly to nonideality, there should be a positive correlation of charge with *k_D_*. Figure 8 shows the correlation of *Z_DHH_* with the diffusion interaction parameter, *k_D_*. Under formulation conditions (Figure 6, panel a) increasing *Z_DHH_* correlates with increased repulsive interaction (i.e., *k_D_* becomes more positive). This suggests that charge measurements may be a useful parameter for selecting candidate mAbs for development. It should be noted that it is the effective charge, *Z_eff_*, rather than *Z_DHH_*, that impacts thermodynamic nonideality [26]. This distinction is important because *Z_eff_* includes the contribution of the solvent ions, with *Z_eff_* decreasing (i.e., repulsive interactions decreasing) as salt concentration is increased [12]. Because salt diminishes charge–charge interactions, thus reducing colloidal stability, it should be no surprise that most mAbs are manufactured and formulated in low-salt solvents.

While charge does contribute to nonideality under formulation conditions, there is no correlation between *Z_DHH_* and *k_D_* under physiological conditions (Figure 8b). This result means that it is unfavorable solvent displacement energies that keep mAbs in solution, for all other protein–protein interactions are attractive [56]. Similarly, it is likely that it is the protein solvation shell that dominates the solubility of serum IgG.

One surprising result of our work is that freshly prepared human IgG exhibits a rather narrow *Z_DHH_* distribution in physiological solvent (from approximately −10 to −2, Figure 9), even though isoelectric focusing shows that the same sample has species ranging from pI < 4 to pI > 10 [14]. This exact same *Z_DHH_* range may be calculated from electrophoretic mobility measurements published 80 years ago [27]. It would seem from these results that IgGs exhibit charge homeostasis. The mechanism for this homeostasis is not clear. None of the mAbs contained anionic carbohydrates (Table 1), so it is not possible to determine whether the addition of, say, sialic acid would result in a more anionic IgG under physiological conditions. Given the narrowness of the human IgG charge distribution under physiological conditions, it seems likely that sialylation contributes specifically to interactions rather than merely impacting the global charge.

Figure 9 shows that most, but not all, of the mAbs in this study exhibit *Z_DHH_* that fall in the range for human serum poly IgG. It is not clear whether there are any physiological or medical consequences associated with a mAb *Z_DHH_* that falls outside the normal physiological range. Thus, these results are presented in the hopes of stimulating further research.

Since both aggregation and high viscosity are reflections of protein–protein interactions, it would be useful to have a rigorous means of determining whether an IgG (or solvent condition) is good, bad or indifferent. We suggest that the apparent thermodynamic nonideality (d*B*_22*-app*_/dC or d*A*_2*-app*_/dC) might fulfill this need. To calculate *A*_2*-app*_, several quantities are required (see Figure 10 legend), but each of these values are tabulated, easily calculated or readily obtained experimentally. A dimer dissociation constant of 1 mM was used to mimic the attractive interactions in all cases. This value of K_d_ is at the upper range of what has been found experimentally [13,15]. If stronger attractive interactions are used (e.g., 300 μM rather than 1 mM), the range where interactions are net attractive is more extensive. A more complete report on determining and using this interaction parameter is being developed.

## 5. Conclusions

Charge is a fundamental property of antibodies and is important in providing colloidally stable mAb solutions during their development, manufacture and formulation. At this time, protein charge cannot be calculated with any accuracy even using the most detailed structural information and the most sophisticated algorithms. Protein charge, however, is readily measured with accuracy and precision. In this first systematic and comprehensive examination of the charge on IgGs it is clear that: (1) IgGs exhibit charge homeostasis in physiological solvent, (2) they appear to bind significant quantities of anions, (3) anion binding will contribute to the desolvation energy, thus preventing IgG aggregation, (4) mAb charge measurements may be useful in selecting candidate molecules for development and (5) mAb charge measurements under physiological conditions may be useful in determining whether a candidate molecule falls within the normal range for human IgGs. 

## Figures and Tables

**Figure 1 antibodies-08-00024-f001:**
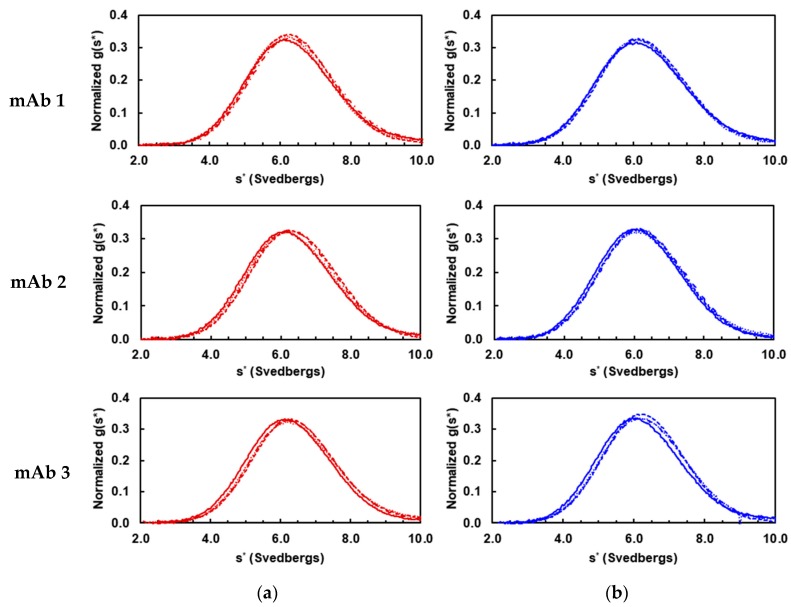
Sedimentation velocity analysis of IgG subclasses from mAb1, mAb2, and mAb3 in (**a**) pH 5.0 acetate (red) and (**b**) pH 7.4 PBS (blue) solutions. Normalized g(s*) sedimentation distributions are obtained from IgG1 (solid line), IgG2 (dotted line), IgG4 (dashed line), and IgG4Pro (dot-dashed line) in both buffers. The curves are superimposed on each other in both panels.

**Figure 2 antibodies-08-00024-f002:**
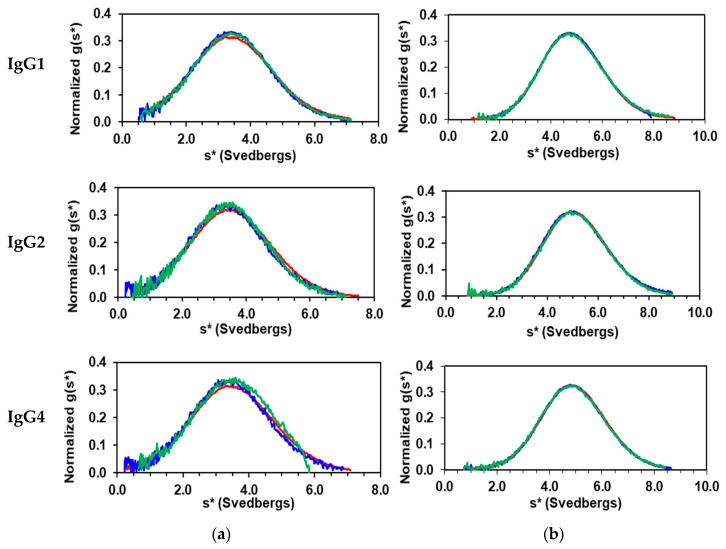
Sedimentation velocity analysis of IgG1, IgG2, and IgG4 cleaved (**a**) Fc and (**b**) F(ab’)_2_ from mAb 1 in pH 5.0 acetate. Normalized g(s*) sedimentation distributions obtained with the concentration of 0.5 mg/mL (red), 0.167 mg/mL (blue), and 0.056 mg/mL (green). The graph for IgG4Pro F(ab)’_2_ is not shown because it is indistinguishable from IgG4. Refer to text for the s_w_ values.

**Figure 3 antibodies-08-00024-f003:**
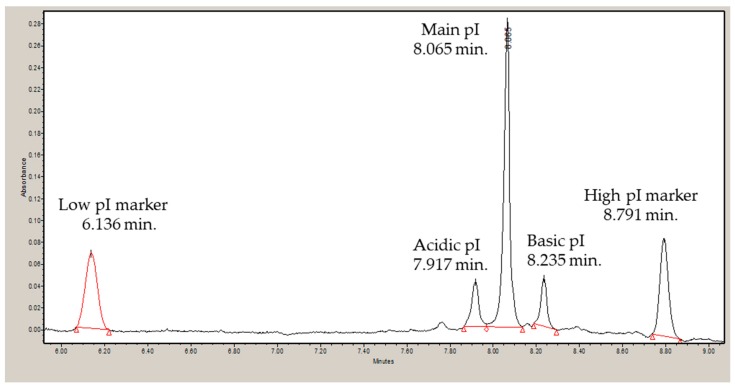
Electrophoretogram image of mAb1 IgG1. The peaks to the left and to the right of the main peak indicates acidic and basic charge variant, respectively.

**Figure 4 antibodies-08-00024-f004:**
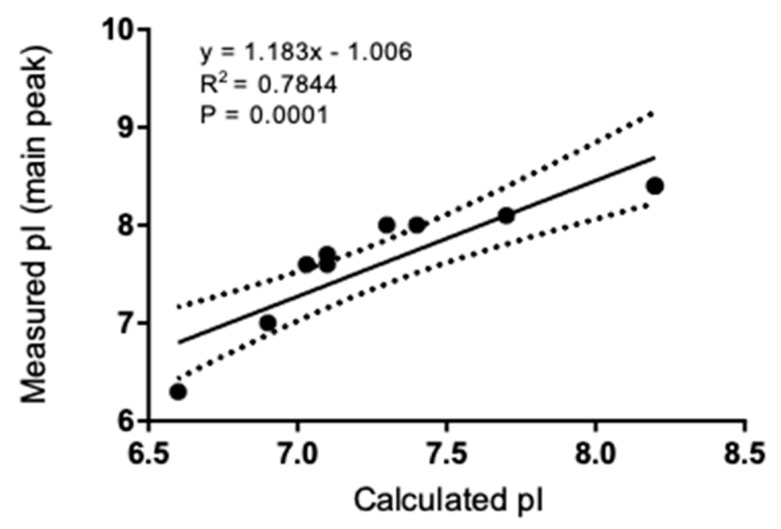
Linear regression analysis and correlation between experimental pI as measured by icIEF and theoretical pI calculated from the IgG sequence. Dotted lines indicate the 95% confidence interval.

**Figure 5 antibodies-08-00024-f005:**
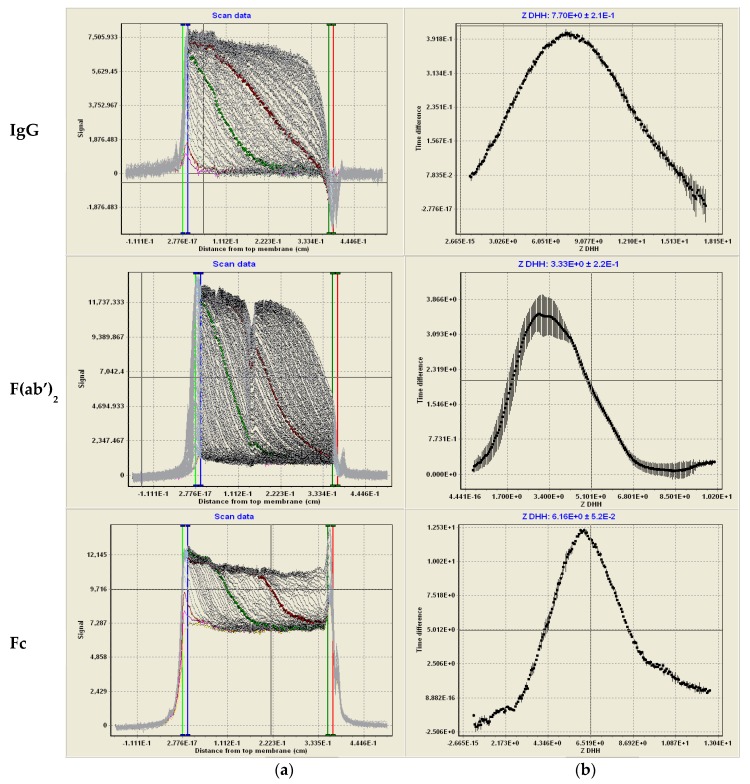
*Z_DHH_* determination of IgG, F(ab’)_2_, and Fc by Membrane-Confined Electrophoresis (MCE) in pH 5.0 acetate. (**a**) Raw MCE scans over time during electrophoresis. The data (left panel) shows the light intensity (I, vertical axis) as a function of the distance moved from the membrane (cm, horizontal). Time difference curves (ΔI/Δt) are calculated from data between the green and red highlighted scans. The electrophoretic mobility distribution is calculated from distance moved from the membrane, x, divided by the product of the electric field, E, and average elapsed time for the middle scan *t*, $\mu =\frac{x}{E\cdot \bar{t}}$. (**b**) The vertical axis shows the time derivative (ΔI/Δt) of the intensity data in panel (**a**) as a function of *Z_DHH_* (horizontal axis). *Z_DHH_* was calculated from the mobility using T = 20 °C; viscosity = 0.98 cp; conductance = 16.8 mS; E = −19.8 V/cm, D = 78; counterion radius, 0.18 nm; Stokes radius, 5.5 nm. The peak *Z_DHH_* position is displayed above the curve.

**Figure 6 antibodies-08-00024-f006:**
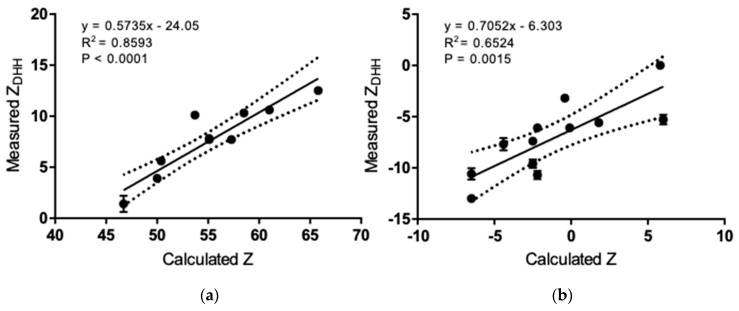
Linear regression analysis and correlation between experimental *Z_DHH_* measured by MCE and theoretical Z calculated from the IgG sequence. (**a**) pH 5.0 acetate. (**b**) pH 7.4 PBS. Dotted lines indicate the 95% confidence interval.

**Figure 7 antibodies-08-00024-f007:**
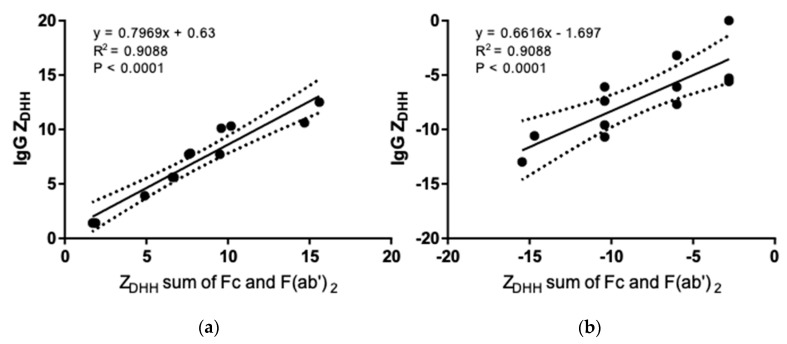
Linear regression analysis and correlation between *Z_DHH_* measured from intact IgG and the sum of *Z_DHH_* from the fragments. (**a**) pH 5.0 acetate. (**b**) pH 7.4 PBS. Dotted lines indicate the 95% confidence interval.

**Figure 8 antibodies-08-00024-f008:**
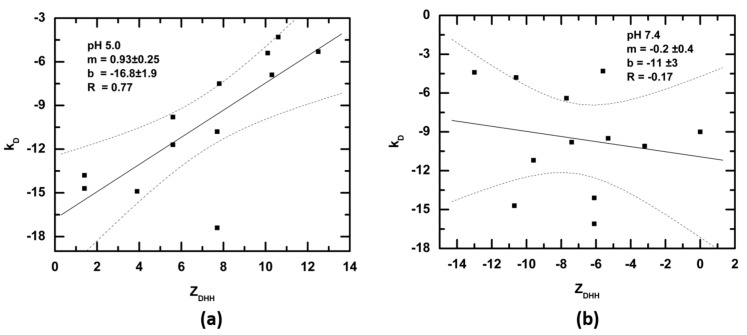
Linear regression analysis and correlation between *Z_DHH_* measured for intact IgG and the concentration-dependence of the diffusion coefficient, k_D_. (**a**) pH 5.0, acetate buffer, (**b**) pH 7.4 PBS. Dotted lines indicate the 95% confidence interval.

**Figure 9 antibodies-08-00024-f009:**
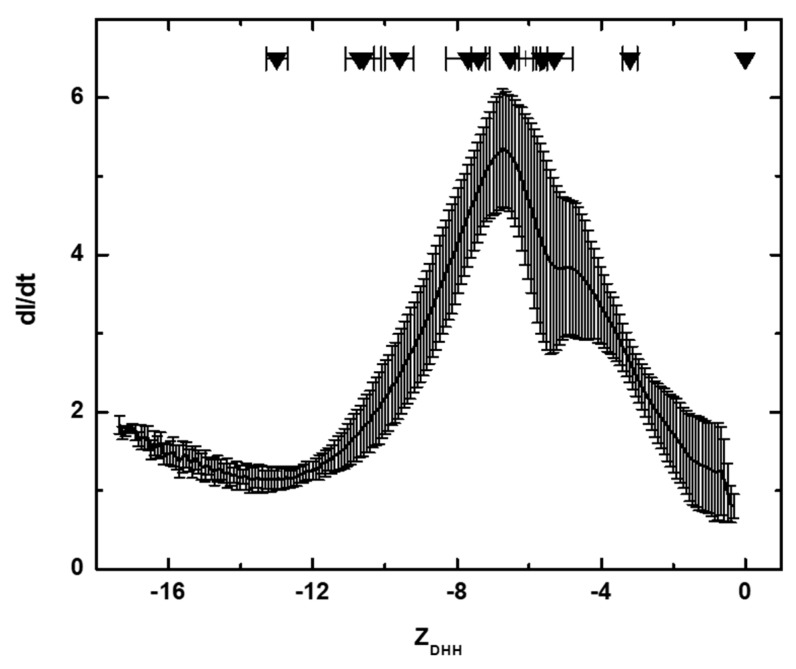
*Z_DHH_* distribution for freshly prepared human IgG in DPBS. *Z_DHH_* was calculated for T = 20 °C, viscosity = 0.98 cp, electric field = −14.88 V/cm, ionic strength = 0.167 M, conductivity = 16.6 ms, protein radius = 5.5 nm, counterion radius = 0.18 nm, D = 78. The *Z_DHH_* for the twelve intact IgGs in this study are noted (inverted triangles) along with bars indicating the measurement uncertainty.

**Figure 10 antibodies-08-00024-f010:**
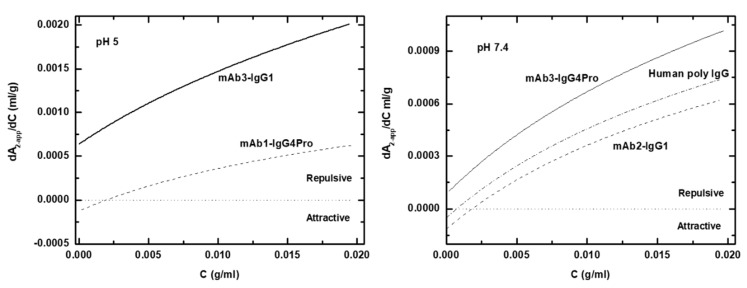
Protein–protein-interaction parameter d*A*_2*-app*_/dC for pH 5 and pH 7.4 data. Note the concentrations are present in g/mL (cgs units) in order to be consistent with the derivation of the equations [12], and correspond to a concentration range of 0–20 mg/mL. The parameters used to generate these curves are: M_1_ = 150,000 g/mole, hydrated radius 4.39 nm, hydration 0.3 g-H_2_O/g-protein, axial ratio 5, monomer–dimer K_d_ 1 × 10^−^^3^ M, protein partial specific volume = 0.73 mL/g, solvent partial specific volume 0.993 mL/g, temperature 20 °C, and solvent density 1.0 g/mL. The salt concentration for pH 5 was set to 60 mMolal, and for pH 7.4 to 150 mMolal. For either condition, curves for the mAbs having the lowest *Z_DHH_* (dashed lines) and highest *Z_DHH_* (solid lines) are shown. For pH 7.4, a curve for human poly IgG (dash-dot) is shown. The horizontal dotted line at 0 corresponds to ideal conditions, with values less than zero corresponding to net attraction and greater than 0 to net repulsion.

**Table 1 antibodies-08-00024-t001:** Evaluation of IgG quality.

ID	Subclass	Monomer (%)	Mass (Da)	Glycoform Level (%) *
G0F 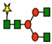	G1F 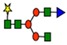	G2F 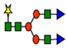
mAb 1	IgG1	>99	148,480	49	43	8
IgG2	>99	147,913	52	39	9
IgG4	>99	148,190	45	39	16
IgG4Pro	>99	148,210	43	40	17
mAb 2	IgG1	>99	148,301	45	42	13
IgG2	>99	147,734	50	41	9
IgG4	>99	148,012	43	43	14
IgG4Pro	>99	148,032	20	50	30
mAb 3	IgG1	>99	149,959	30	52	18
IgG2	>99	149,231	45	41	14
IgG4	>99	149,507	49	43	8
IgG4Pro	>99	149,529	25	52	25

* N-acetylglucosamine ■; mannose ●; galactose ►; 
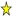
 fucose.

**Table 2 antibodies-08-00024-t002:** Quality of IgG fragments from IdeS digestion.

Subclass	V Region	Cleaved Site	F(ab’)_2_ Purity (%)	Fc Purity (%)
IgG1	mAb 1	CPPCPAPELLG/GPSVF	95	100
	mAb 2		100	
	mAb 3		100	
IgG2	mAb 1	CPPCPAPPVA/GPSVF	100	98
	mAb 2		100	
	mAb 3		100	
IgG4	mAb 1	CPSCPAPELLG/GPSVF	95	97 *
	mAb 2		95	
	mAb 3		97	
IgG4Pro	mAb 1	CPPCPAPELLG/GPSVF	97	
	mAb 2		100	97 *
	mAb 3		100	

* The cleaved Fc is identical between IgG4Pro and IgG4 because the enzymatic digest occurred below the hinge region.

**Table 3 antibodies-08-00024-t003:** Measured and calculated pI values of IgG.

ID	Subclass	pI_cal_	pI_icIEF_		
			Acidic Peak	Main Peak	Basic Peak
mAb 1	IgG1	7.7	7.9	8.1	8.2
	IgG2	6.9	6.9	7.0	7.3
	IgG4	6.6	6.2	6.3	6.5
	IgG4Pro	6.6	6.2	6.3	6.5
mAb 2	IgG1	8.2	8.2	8.4	8.6
	IgG2	7.3	7.9	8.0	8.2
	IgG4	7.0	7.4	7.6	7.7
	IgG4Pro	7.0	7.4	7.6	7.7
mAb 3	IgG1	8.2	8.2	8.4	8.6
	IgG2	7.4	7.2	8.0	8.1
	IgG4	7.1	7.5	7.7	7.8
	IgG4Pro	7.1	7.5	7.7	7.8

**Table 4 antibodies-08-00024-t004:** Measured and calculated Z values of IgG, F(ab’)_2_, and Fc in pH 5.0 acetate.

ID	Subclass	IgG		F(ab’)_2_		Fc *	
		*Z_DHH_*	*Z_cal_*	*Z_DHH_*	*Z_cal_*	*Z_DHH_*	*Z_cal_*
mAb 1	IgG1	7.7 ± 0.2	57.3	3.3 ± 0.2	31.2	6.2 ± 0.1	26.30
	IgG2	3.9 ± 0.1	50.0	0	25.9	4.9 ± 0.1	24.30
	IgG4	1.4 ± 0.2	46.7	1.3 ± 0.1	27.9	0.45 ± 0.1	18.98
	IgG4Pro	1.4 ± 0.8	46.7	1.5 ± 0.2	27.9	0.45 ± 0.1	18.98
mAb 2	IgG1	10.6 ± 0.1	61.0	8.6 ± 0.2	34.9	6.2 ± 0.1	26.30
	IgG2	10.1 ± 0.2	53.7	4.7 ± 0.1	29.6	4.9 ± 0.1	24.30
	IgG4	5.6 ± 0.2	50.4	6.2 ± 0.1	31.6	0.45 ± 0.1	18.98
	IgG4Pro	5.6 ± 0.2	50.4	6.2 ± 0.1	31.6	0.45 ± 0.1	18.98
mAb 3	IgG1	12.5 ± 0.1	65.8	9.4 ± 0.1	39.6	6.2 ± 0.1	26.30
	IgG2	10.3 ± 0.2	58.5	5.3 ± 0.2	34.3	4.9 ± 0.1	24.30
	IgG4	7.7 ± 0.2	55.1	7.1 ± 0.1	36.3	0.45 ± 0.1	18.98
	IgG4Pro	7.8 ± 0.2	55.1	7.3 ± 0.1	36.3	0.45 ± 0.1	18.98

* The value from each subclass is identical across the mAb set because it was measured on pooled Fc samples from the three mAb digestions.

**Table 5 antibodies-08-00024-t005:** Measured and calculated Z values of IgG, F(ab’)_2_, and Fc in pH 7.4 PBS.

ID	Subclass	IgG		F(ab’)_2_		Fc *	
		*Z_DHH_*	*Z_cal_*	*Z_DHH_*	*Z_cal_*	*Z_DHH_*	*Z_cal_*
mAb 1	IgG1	−5.6 ± 0.1	1.8	0	−0.48	−2.8 ± 0.1	1.50
	IgG2	−7.7 ± 0.6	−4.4	0	−4.59	−6.0 ± 0.6	−0.48
	IgG4	−10.6 ± 0.5	−6.5	−4.3 ± 0.8	−2.61	−10.4 ± 0.3	−4.60
	IgG4Pro	−13 ± 0.3	−6.5	−5.05 ± 0.5	−2.61	−10.4 ± 0.3	−4.60
mAb 2	IgG1	0	5.8	0	3.5	−2.8 ± 0.1	1.50
	IgG2	−3.2 ± 0.2	−0.4	0	−0.61	−6.0 ± 0.6	−0.48
	IgG4	−7.4 ± 0.2	−2.5	0	1.38	−10.4 ± 0.3	−4.60
	IgG4Pro	−9.6 ± 0.4	−2.5	0	1.38	−10.4 ± 0.3	−4.60
mAb 3	IgG1	−5.3 ± 0.5	6.0	0	3.45	−2.8 ± 0.1	1.50
	IgG2	−6.1 ± 0.3	−0.1	0	−0.36	−6.0 ± 0.6	−0.48
	IgG4	−6.1 ± 0.2	−2.2	0	1.63	−10.4 ± 0.3	−4.60
	IgG4Pro	−10.7 ± 0.4	−2.2	0	1.63	−10.4 ± 0.3	−4.60

* The value from each subclass is identical across the mAb set because it was measured on pooled Fc samples from the three mAb digestions.

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
