# Peer review of "IgG Charge: Practical and Biological Implications"

_2073-4468, 2019, doi:10.3390/antib8010024_

Reviewer 1 Report

The paper shows the studies done with different subclasses of IgG by measuring the influence of charge on thermodynamic properties of the protein. The authors have done a good job in their investigations and explained the data precisely.

A few points need further attention:   

1.     The title says ‘biological consequences’ which does not seem to be appropriate as there is no biological investigation done in the present study. Therefore it is suggested to modify the title according to the content of the paper.

2.     Introduction and background convey the same idea and can be combined. 

3.     Figures 3 and 5 need more clarity. The scale bars are not legible.  

Author Response

1.       We changed “consequence” to “implications”. While mAb charge is mainly used to understand HCLF in storage buffers during mAb product development in the industry, this study instead aimed to investigate the mAb charge in physiological conditions and compare it to endogenous IgGs derived from human to shed light on its role in biological activities. Although there is no direct biological study performed, the objective is to enhance this understanding.

2.       We changed “Background” to “Theoretical basics of protein charge”. The “Introduction” contains information on the significance of charge on product development and biological processes, while the now changed sub-title “Theoretical basics of protein charge” describes the theory and fundamentals of how protein charge is calculated.

3.       Due to a recent job change, I cannot access the computer and database where the original data were generated and stored in order to make direct changes to the legend size and color, etc. on the instrument computer. For Figure 3, we listed the retention times on top of the peaks to provide clarity. For Figure 5, we described in the legend what the x- and y- axis represent. There is a higher resolution of each of the Figures provided to the publisher. We will inform the publisher to make the scale bars more bold and legible during post-production. 

Reviewer 2 Report

Authors made all the suggested in first revision. Now it looks much better

Author Response

Reviewer comment: "Authors made all the suggested in first revision. Now it looks much better"

Review didn't suggest any more changes to be made. 

This manuscript is a resubmission of an earlier submission. The following is a list of the peer review reports and author responses from that submission.

Round  1

Reviewer 1 Report

The authors have reported an interesting study by investigating the charge on antibodies. They employed different techniques for this purpose and reported ZDHHas one of the reliable methods. They compared the charged state of antibodies in different conditions such as pH 5 and 7. They also showed the charge distribution on Fc and F(ab’)2 fragments. Their data indicated large differences in the values indicating that charge distribution may be explored further to check for antibody formulations.

Some additional points which need to be addressed are as follows: 

They have not talked about the issues that are affected by changes in the charge of the antibodies such as PK, blood clearance and efficacy with examples to emphasize the importance of the issue. Very few references have been given from the previous studies. Other reports of the antibody charge investigations need to be referenced and discussed e.g (https://doi.org/10.1080/19420862.2016.1225642). Researchers have reported acidic, main and basic antibodies previously which have not been talked about. 

Authors say ‘It is known that charge and charge distribution are important contributors to protein solubility and solution viscosity’ ‘Protein charge is significant to a variety of biochemical, biophysical and biological phenomena’. Here, examples would be helpful. Also, if there are reports of some problems associated with antibodies due to this issue? 

Furthermore, authors say ‘It is apparent that a systematic analysis of the charge on mAbs would be useful.’ How so? 

Figure 1 needs to be represented in clarity. The peaks are not clearly visible. 

Figure 2: Why IgG4 Pro not shown here?

Figures 3 and 5: The background needs to be removed to clearly show the data. 

Figure 8: a and b not indicated in the figure legend.

Table 3 and 4: Missing information in Fc column for mAb2 and mAb3

References needed at several places e.g. ‘These data are consistent with a model in which an anion is bound for every 1.3 – 2 H+ bound’

Discussion: The authors say, ‘Because protein charge can vary with solvent conditions, it is a system property rather than a property of the protein’. This point needs elaboration. 

Conclusion: The authors write, ‘At this time, protein charge cannot be calculated with any accuracy by even the most detailed structural information using the most sophisticated algorithms. Protein charge, however, is readily measured with accuracy and precision.’ These are two opposite statements? What do they mean with this?

Reviewer 2 Report

Overall this article hypothesis is very interesting and the need of the hour with a growing interest in the antibody structure. However, this manuscript falls short of interpretation of results. Here is some of the major revisions are required prior to publishing.

1.   Authors made a good attempt to understand therapeutic antibody charge using analytical electrophoresis technique. 

2.   Figure 1 graph colors look bit confusing, especially blue dotted line and solid line are not distinguishable.  

3.   Figure 3 and Figure 5 axis title font is very small. These two figures are not in journal format.

4.   Results are very interesting but the representation of results is very confusing. It is highly recommended to use subtopics like methods section to separate different results. 

This would strengthen manuscript and helps readers to understand.